# Social Priming in Speech Perception: Revisiting Kangaroo/Kiwi Priming in New Zealand English

**DOI:** 10.3390/brainsci12060684

**Published:** 2022-05-24

**Authors:** Gia Hurring, Jennifer Hay, Katie Drager, Ryan Podlubny, Laura Manhire, Alix Ellis

**Affiliations:** 1New Zealand Institute of Language, Brain, and Behaviour, University of Canterbury, 20 Kirkwood Avenue, Upper Riccarton, Christchurch 8041, New Zealand; jen.hay@canterbury.ac.nz; 2Department of Linguistics, University of Canterbury, 20 Kirkwood Avenue, Upper Riccarton, Christchurch 8041, New Zealand; lauraman22@gmail.com (L.M.); rory.alix.ellis@gmail.com (A.E.); 3Department of Linguistics, University of Hawai‘i at Mānoa, 1890 East-West Rd, Honolulu, HI 96822, USA; kdrager@hawaii.edu; 4Department of Linguistics, University of Alberta, Edmonton, AB T6G 2G4, Canada; pudplace@gmail.com

**Keywords:** priming, speech perception, sociophonetics, lexical decision task, New Zealand English, Australian English

## Abstract

We investigate whether regionally-associated primes can affect speech perception in two lexical decision tasks in which New Zealand listeners were exposed to an Australian prime (a kangaroo), a New Zealand prime (a kiwi), and/or a control animal (a horse). The target stimuli involve ambiguous vowels, embedded in a frame that would result in a real word with a KIT or a DRESS vowel and a nonsense word with the alternative vowel; thus, lexical decision responses can reveal which vowel was heard. Our pre-registered design predicted that exposure to the kangaroo would elicit more KIT-consistent responses than exposure to the kiwi. Both experiments showed significant priming effects in which the kangaroo elicited more KIT-consistent responses than the kiwi. The particular locus and details of these effects differed across experiments and participants. Taken together, the experiments reinforce the finding that regionally-associated primes can affect speech perception, but also suggest that the effects are sensitive to experimental design, stimulus acoustics, and individuals’ production and past experience.

## 1. Introduction

Can priming with regionally-associated images affect speech perception? Previous work suggests that it can [1]. New Zealanders primed with kangaroos appeared to shift their vowel perception to be more Australian-like. However, the task used in that experiment was not unambiguously about speech perception. Our paper adopts a more controlled, preregistered design, which more directly tests whether priming with regionally-associated images can lead to shifts in perception.

Across two experiments, participants are primed with images of kangaroos, kiwis, and horses, while conducting a lexical decision task with words containing a vowel that could be categorized differently in New Zealand English and Australian English. Both experiments find some evidence that the regional prime can affect what vowel is heard. Overall, the results reinforce the claim that regionally-associated primes can affect speech perception, but also provide some reason for caution—suggesting that such effects are sensitive to aspects of experimental design, stimulus acoustics, and individual differences.

## 2. Background

### 2.1. Linguistic Terminology

This paper, like the literature it builds on, uses lexical sets to refer to the vowels under investigation [2]. Lexical set terminology provides a dialect-neutral way of referring to classes of vowels, as opposed to, for example, using the international phonetic alphabet, which references the production of a specific token of a vowel but does not identify (in any dialect-neutral way) which class of vowel is being discussed. For example, in NZE and Australian English, the production of the vowel in the word ‘fish’ would receive quite different phonetic transcriptions, but in both dialects the vowel belongs to the KIT lexical set, and thus shares its production with a set of other KIT words (such as *bit*, *miss*, *him*, etc.). This paper is focused on vowels in the KIT lexical set, and also the DRESS lexical set—namely, words that share a vowel with ‘DRESS’ (such as *bet*, *mess*, and *hem*). 

### 2.2. Social Priming in Speech Perception

An array of language external factors has been shown to influence speech perception. Among such factors is social information attributed to the talker, including gender [3,4], age [5,6], ethnicity [7], sexual orientation [8], socioeconomic status [5], region [9,10], attractiveness [11], and social persona [12]. Most of this work either uses photographs or videos paired with different talkers to manipulate perceived characteristics of the talker (e.g., [3,5]) or else manipulates listeners’ expectations through explicit descriptions of the talkers and their characteristics (e.g., [9,12]). 

Understanding social priming in listener perception is important because it influences the relevant scope of speech perception models. What is the extent to which non–acoustic information is integrated in the process of speech perception? And is the creation and interpretation of social meaning an integral part of the speech perception process, or is it a separate process which does not influence speech perception at all? These questions have been the topic of debate regarding the modularity of speech perception, leading to the emergence of models in which social factors play an important role in speech perception (see [13,14,15]). The reported social priming results have fed directly into that discussion (see [15,16,17,18,19,20,21,22]). We follow previous works, using the term ‘speech perception’ to describe how listeners perceive the acoustic signal. More precisely—in this paper we will be concerned with which phoneme/word an acoustic signal is mapped onto. A reviewer asks us to consider whether our results relate to ‘perception’ or a later stage of ‘interpretation’ (following [23]). We take the mapping of acoustic input to linguistic categories to be a fundamental part of ‘speech perception’, broadly construed. However, we note here that by using this term, we are not making any claim about the specific stages of processing that are involved in this mapping.

One key social priming study adopted a paradigm in which participants listen to a sentence and then match the realization of a target word from the sentence with one from a synthesized vowel continuum [9]. Niedzielski (1999) [9] showed that listeners from Detroit matched a target with a raised variant of a diphthong when they thought they were listening to a Canadian, but not when they thought the speaker was from Michigan. These findings were interpreted as showing that regional expectations about where a speaker was from could affect speech perception. The same task was subsequently employed in New Zealand, in a task in which half the listeners had ‘Australian’ written on the answer-sheet, and half had ‘New Zealander’ [10]. Those participants with Australian matched KIT vowels to more raised variants, typical of Australian English. The effect was present for women, but not men, although the gender ratio in the sample was not balanced. The participants were surveyed at the end of the experiment about where they thought the speaker was from, and nearly all participants responded that they thought the speaker was from New Zealand. This finding led the authors to speculate that the effect was not driven by expectations or beliefs about the speaker, but perhaps by a more automatic priming effect.

In Hay & Drager (2010) [1], we, therefore, tested this possibility explicitly through employing the same task as Hay, Nolan & Drager (2006) [10] but exposed participants to incidental social primes associated with the two regions. To achieve incidental priming, the experimenter pulled out from a cabinet one of two sets of stuffed animal toys prior to beginning the experiment, pretending like she did not know why they were there and setting them aside but within view of the participant. The first set of stuffed toys were kangaroos and koalas associated with Australia, and the second set were kiwis associated with New Zealand. The results from this second study were remarkably similar to those observed in the original study—including a gender difference. The presence of the regionally-associated stuffed toys—items that the participants were led to believe had nothing to do with the task—appeared to have influenced vowel perception in much the same way as regional labels.

Speculating that the observed difference between men and women may ultimately stem from attitudinal differences, Walker et al. (2018) [24] conducted the same task with an attitudinal prime, in which participants conducted a baseline condition, and then were exposed to one of three sets of facts about Australia (good, neutral, vs. bad facts). The manipulation shifted their performance in the subsequent task, and the authors concluded that perceptual adaptation towards a dialect can occur in the absence of a speaker of that dialect, and that these adaptations can be subject to a listener’s affect towards the primed dialect region.

These experiments are all from New Zealand, and it is important to note that priming with regionally-associated images has not been replicated outside of New Zealand. In an experiment conducted in Australia, Walker, Szakay and Cox (2019) [25] found that Australian participants were not influenced by exposure to the animals. They suggest that the lack of an effect may be due to differences between Australians’ and New Zealanders’ metalinguistic awareness of the relevant variation (p. 21). 

Attempts to replicate the general design from Niedzielski (1999) [9] in other dialect areas have also been mixed. For example, Jannedy et al. (2011) [26] showed that written dialect areas on an answer sheet (following [9]) significantly shifted perceptions of German fricatives. Whereas Lawrence (2015) [27] used a similar design and found limited evidence for social priming of BATH/STRUT vowels in speakers of Standard Southern British English.

While we do not necessarily expect these effects to replicate in places with different language experiences and stereotypes, the mixed results raise questions about the validity of the findings presented in Hay & Drager (2010) [1]. Did the results presented therein arise due to chance, or is the lack of a finding in other work due to, for example, differences in either exposure to or salience of sociolinguistic variation for the community of participants tested? 

In addition to the failed replication attempts outside of New Zealand, the very task itself raises questions about how exactly to interpret the results. The above experiments all involve the same task (in which participants hear a vowel embedded within a sentence and are then asked to match it to the closest token on a synthesized continuum). This task is unnatural in that the process involves holding the target vowel in memory, while participants are exposed to other realizations of the same vowel and then perform a matching task. These effects might be attributed to memory then and are thus not unambiguously a meaningful part of speech perception. Because the task does not allow for an immediate response, we cannot be sure that the regionally-associated primes have actually influenced the perception of the target vowel. Such a task incorrectly presumes that memory does not degrade over time and that it is not influenced by the presentation of subsequent auditory input. Therefore, even if the primes were effective, and the observed effect was not due to chance, the primes may have influenced the selection of tokens from the vowel continua (i.e., a process downstream from the initial recognition and mapping) instead of influencing perception per se. 

In the current paper we are primarily concerned with the priming of social information, whereby socially charged information that is deemed incidental to the speech signal influences listener-behaviour (see [28], to appear, for a discussion). This automatic social priming would thus exclude effects that stem from overtly manipulating expectations by describing characteristics of the talker, and it also excludes studies that manipulate social information attributed to a talker through the use of photographs or video because they may arise due to multisensory or multimodal integration (e.g., [29]) rather than priming per se. Therefore, we ask: does exposure to social information influence the perception of sounds even when the social information is believed to be incidental to the talker, or the language forms they produce? One study in this literature [30] uses something incidental—listener location—with a task that is simpler and more directly related to speech perception. The incidental prime is not explicitly social, but rather—relates to previous experience in different locations. In a simple listening experiment, listeners listen to tokens on a HEAD-HAD continuum and identify what word they hear. Listeners who first conduct the listening experiment while sitting in a car, have a different threshold between DRESS and TRAP than those who first complete it in the lab. They claim that this result—more unambiguously about speech perception—is likely to arise from the same automatic mechanism that elicited the priming by the kangaroos and kiwis in Hay & Drager (2010) [1].

The current study addresses the methodological concerns by attempting to replicate the social priming reported in Hay & Drager (2010) [1] using a completely different experimental paradigm. Using a modified lexical decision task, we examine the extent to which drawings of kangaroos and kiwis may influence vowel perception. We do this by creating ambiguous vowels that are likely to be heard as ‘KIT’ in Australia and ‘DRESS’ in New Zealand. By embedding the vowels in different lexical frames, we can use responses in a lexical decision task to establish what was heard. If we embed an ambiguous vowel (X) in the frame fXzzy, for example, then a ‘yes’ response indicates that the KIT vowel was likely heard (i.e., real word *fizzy*), whereas a ‘no’ response would indicate that DRESS was heard (i.e., nonsense word *fezzy*).

### 2.3. Australian and New Zealand English Vowels

New Zealanders’ and Australians’ realizations of the front vowels are largely distinct. Figure 1 shows the vowel spaces for a number of dialects of English, recorded as stimuli for a perception experiment [31]. As shown in Figure 1, TRAP and DRESS vowels are both realized higher in F1-F2 space in New Zealand English (NZE) compared to Australian English (AusE), with DRESS overlapping with FLEECE for at least some NZE speakers. In AusE, KIT overlaps with FLEECE, whereas it is realized in a much more central position in NZE. Overlap of DRESS or KIT with FLEECE is not a feature that is present in the other dialects studied by Shaw et al. (2018) [31]. Our dialects of interest, then, are distinct from other varieties in having a very high front short vowel and can be differentiated from each other with respect to the specific identity of that vowel.

Due to the high front position of KIT in AusE and DRESS in NZE, it seems likely that a fairly high front short vowel in a lexically ambiguous context is likely to be perceived as KIT by Australians, and DRESS by New Zealanders. Indeed, by analyzing error patterns from the Shaw et al. (2018) [31] experiment, we can examine what happens when a New Zealander hears an Australian KIT vowel in isolation. The authors played recordings of words from NZE and AusE to listeners in each country. In some cases, these words were in isolation, and in some cases after a period of exposure to a speaker from one of the dialects reading a story. Listeners identified what vowel they heard in the word. We examined the patterns of response errors in the data from that experiment, to confirm that there is variation in New Zealand listeners’ perceptions of KIT and DRESS. In particular, an Australian KIT vowel is more likely to be heard as DRESS than KIT, but the likelihood of identifying it accurately as KIT substantially increases for listeners who were exposed to an Australian voice prior to hearing the target word. New Zealanders’ perception of DRESS, on the other hand, is more stable and more often heard as DRESS than any other phoneme, regardless of the dialect it is produced in.

The acoustic configuration of the DRESS and KIT vowels, together with this observed pattern of errors, suggests that it should be possible to create stimuli that are ambiguous between a NZE DRESS and an AusE KIT, and that it might be possible to influence the mapping of these signals during speech processing using regionally-associated primes. 

For the two experiments presented herein, we created a set of words with synthesized vowels with F1 and F2 positioned between the NZE and AusE KIT acoustic spaces. For New Zealand listeners, the vowel could therefore be heard as either a NZE KIT or an AusE KIT; and it could also be mapped to NZ DRESS, as in the Shaw results described above. We hypothesize that listeners are unlikely to classify the vowel as DRESS if they are primed towards Australia, and more likely to classify it as DRESS if they are primed toward NZ.

The experiments use a lexical decision task so that we can infer from participants’ responses which vowel they heard. The frames for the vowels form a real word context for only one of the vowels (i.e., either DRESS or KIT) and not the alternative vowel. For example, if a listener perceived the ambiguous vowel (X) as KIT, they would interpret stimuli such as tXpping and fXzzy as real words and scXptic and pXppered as nonsense words. Alternatively, the opposite would be true if they mapped the vowel to DRESS. We refer to contexts where a KIT vowel results in a real word as being in a ‘KIT frame’ and contexts where a DRESS vowel results in a real word as being in a ‘DRESS frame’. Central to the present work, priming listeners with different, culturally charged animal images allows us to test whether use of a Kangaroo prime (associated with Australia) will lead New Zealand listeners to hear the ambiguous vowel as KIT rather than DRESS (i.e., to respond in a more Australian-like way).

We preregistered a design, which is reported here as experiment one (Section 3). Experiment two (Section 4) adjusts various elements of the experimental design and repeats the experiment.

## 3. Experiment One

We used a lexical decision task to test our hypotheses that (1) exposure to regional primes would affect vowel perception in ways that are consistent with the relevant regional dialects and (2) that the effect of the primes is not solely due to listener expectations about where the talker is from.

### 3.1. Materials and Methods

The experiment was run on the internet. Participants completed a lexical decision task while images were presented on the screen. Responses to target words were to an ambiguous vowel embedded in one of two **frames** (a **KIT-frame** or a **DRESS-frame**).

We used line drawings of kangaroos, kiwis and horses as different **prime-types**. The prime-types were presented in one of two **presentation-types**—either as if they were the character talking the words (the **speaking** presentation-type), or just incidentally on the screen, presenting some instructions (the **incidental** presentation-type). The speaking presentation-type tests the effect of expectations or beliefs about the speaker (to the extent that a participant believes that the animal is ‘speaking’) in line with Hay, Nolan & Drager (2006) [10]. The incidental presentation-type tests the effect of incidental exposure to regional primes, in line with Hay & Drager 2010 [1].

Each participant was selected either into the baseline condition or the priming condition, and into either a speaking or incidental presentation-type. Each participant responded to words across three blocks, each of which used a different **voice** and used different images for each block. Participants in the **baseline condition** encountered pictures of three different coloured horses in the three blocks, presented as either speaking (for participants in the speaking presentation-type) or presented incidentally. Participants in the **priming condition** encountered pictures of a horse, a kangaroo and a kiwi, again either speaking or incidentally. More detail on all manipulations is given below.

#### 3.1.1. Auditory Stimuli

Target items contained an ambiguous vowel embedded in one of two **frames:** a KIT-frame or a DRESS-frame. **KIT-frames** are items that are real words if they contain KIT but nonsense words if they contain DRESS, whereas the opposite is true for **DRESS-frames**. Filler items contained either LOT or STRUT. Example frames for target and filler items are shown in Table 1. A complete list of the frames used can be found in Appendix B.

Stimuli are all trochaic, with the target vowel in the first syllable. Half of the stimuli (90 items) contained the ambiguous vowel, half of them in a KIT frame (where only if the vowel was heard as KIT would the stimulus be a real word), and half of them in a DRESS frame. Fillers were designed so that half of the fillers would be heard as real words, and half would be heard as non-words. The fillers included words with STRUT and LOT, neither of which are likely to have led to major misunderstandings across dialects. Also, as fillers, we also included items which the listener would hear as real words regardless of which vowel was heard (**the Both-frame**—e.g., bXgger,), and which neither vowel would make the frame a real word (**the Neither-frame**—e.g., kXzzard). 

The target stimuli (KIT-frame and DRESS-frame) have similar CELEX wordform frequencies [32]. The summary statistics are depicted in Table 2. Words with more than one CELEX entry were summed by their wordform and those with zero frequencies were included. All words have low frequency, and a Wilcoxon rank-sum test returned a *p*-value of 0.25, suggesting that the two stimuli frames are not significantly different from each other. The highest frequency words in each frame (KIT-frame and DRESS-frame, respectively) were *giving* and *spending*, while the lowest frequency words (with zero frequency wordforms in CELEX) were *stingers* and *sketchers* (KIT-frame and DRESS-frame, respectively).

#### 3.1.2. Stimuli Recording and Vowel Resynthesis

The ambiguous vowels were created by taking recordings of two voices producing the target KIT word—a New Zealand voice and an Australian voice—and using these voices to synthesize a stimulus that was intermediate between the two vowels.

In stimuli generation, our goal was to synthesize a realistic sounding, stepwise progression that spans true New Zealand and Australian KIT vowels at either extreme. For example, formant values associated with the vocalic portion of a given New Zealand syllable would be manipulated systematically, and incrementally throughout the intermediary steps to become more like that vowel’s Australian counterpart until the actual AusE vowel is incorporated as the final step. To create stimuli in different voices, to span our different blocks, we created three continua for each stimulus item, by recording three NZE speakers, and mixing them each together with a single AusE speaker.

Four female speakers were recruited with both age and height in mind, aiming to reduce inter-stimulus differences that might be rooted in physiology. Thus, three speakers of NZE (mean age = 23 years, mean height = 174.6 cm) and one speaker of AusE (age = 32 years, height = 177 cm) were recorded producing wordlists that included both the target words (KIT-variant) and filler items. For labelling purposes, the three speakers from New Zealand are differentiated as NZ1, NZ2, and NZ3 and the Australian speaker is identified as AU. Recordings were captured in a sound-attenuated booth on the University of Canterbury campus using a Beyerdynamic Opus 55.18 MK II head-mounted condenser microphone, and a digital VU meter to identify and compensate for differences in speaker loudness. Signals were routed through a Sound Devices USBPre 2 audio interface and recorded as WAV files on a late-2013 Macbook Pro laptop computer via Praat [33] at a sampling rate of 44.1 kHz and bit-depth of 16. In order to use the *SpeechInNoise* tool to run the experiment (see Section 3.1.3), which best suited our needs for an online presentation of the experiment, it was necessary to later downsample these source stimuli to 22,050 Hz and convert them to MP3. The MP3 format was required by the platform, and downsampling the stimuli dramatically reduced stimuli load times for participants; piLOT tests incorporating the 44.1 kHz stimuli had problematically long wait times that participants sometimes misinterpreted as crashes/errors.

The creation of continua for each item, for each speaker, was automated using a Praat script authored by Winn (2014) [34] which involves a hybrid of parametric and concatenative synthesis (this script can be accessed at: http://www.mattwinn.com/praat/Make_Formant_Continuum_v30.txt, accessed on 7 April 2017). The script allows users to independently isolate a target time range in each of two sound files, which serve as the bases for the parametric synthesis; the script also excises segments immediately preceding and following each target for appropriate concatenation following the generation of each synthetic target. Original intensity contours are retained. All steps in a given continuum are framed by the same preceding and following segments, although the user specifies from which file those segments are taken. For example, a target removed from the middle of hypothetical “SoundNZ” could serve as one extreme of the continuum, whereas another target removed from the middle of hypothetical “SoundAus” would serve as the basis for the other extreme. The script provides a form that allows the user to set pertinent parameters to values that suit their needs before creating the designated number of steps, interpolating formant values for F1-F4 every 40 ms. The script then outputs sound files for each step where all steps are framed by the preceding and following segments from “SoundNZ” or “SoundAus” as directed. The following parameter settings resulted in satisfactory outputs for our needs: we elected to have five steps per continuum; specified Praat should recognise 4 formants within the specified pitch range (with the exception of targets with adjacent nasals, in which case we specified 5); maximum formant frequency was set to 5000 Hz (with the exception of targets with adjacent nasals, in which case we specified 6000); and the output file intensity was set to normalise all segments and steps to an amplitude of 73 dB. All other parameters retained their default settings. All told, three continua were generated for each stimulus item, pairing each of NZ1, NZ2, and NZ3 with AU.

Preliminary informal testing indicated the third continuum point was perceived by listeners to be ambiguous between an Australian KIT and New Zealand DRESS, and this step was selected for use in experiment one (see Figure 2 and Table 3). Thus, the vowel in experiment one is not exactly like any vowel from NZE or AusE. It is halfway between a NZE KIT and an Australian KIT (and thus also approximately equidistant between a NZE DRESS and an NZE KIT). It is not near an Australian DRESS. Furthermore, the analysis revealed that these vowels were not dissimilar in length and remain around the same length in both these recorded word lists and in a corpus of NZE natural conversation speech (for NZE KIT and DRESS). Having the vowel lengths near equal minimizes the likelihood for a listener to be able to distinguish these vowels by their length. Thus, if primed with an Australian image, we would expect fewer DRESS-like responses to the stimulus, and more KIT-like responses whereas a New Zealand image might be expected to produce a more balanced distribution between KIT/DRESS.

We attempted to synthesize a larger number of stimuli than shown in Table 1, and the final number of words was reduced to words whose synthesis was successful across all three speakers. The range of formant values for an example continuum are depicted in Table 3. 

#### 3.1.3. Conditions and Prime Type

Three cartoon images were commissioned for use as visual primes in this study, all composed by a single artist (Andrew Kepple, see an overview of Kepple’s work here: https://en-academic.com/dic.nsf/enwiki/2553002, accessed on 31 March 2018). These different animals will be referred to as **prime-types**. The kiwi is a prevalent national emblem of New Zealand, so was included to prime listeners for NZE. Similarly, the kangaroo is a prevalent national emblem of Australia and was selected for priming AusE. An image of a horse was included as a ‘neutral’ or baseline prime (see Figure 3). Thus, these animals were selected for their cultural significance in Australasia, or in the context of the horse for its relative neutrality. Additionally, the use of the kiwi and kangaroo make for a more ready comparison to the primes used by Hay & Drager (2010) [1]. We made two additional versions of the horse by manipulating the colour and the aspect ratio of the image.

The four conditions were presented as follows in Figure 4a–d (note the black arrow indicates participants clicking ‘next’ to start listening to the stimuli). Participants were in a baseline-speaking, baseline-incidental, priming-speaking or priming-incidental condition. Each block had an initial instruction screen, and then an experiment screen that remained visible during the block. Each screen contained an animal and a stick figure. In the speaking condition, the animal is speaking, and the stick figure delivers instructions. In the incidental condition, the stick figure is speaking, and the animal delivers instructions. The instruction-giver remains present on the screen throughout the block.

#### 3.1.4. Online Word Recognition Task

The experiment was designed using a *Speech In Noise 2* platform, developed by Chan (2015) [35] (this documentation can be accessed at: https://northwestern.app.box.com/s/9g2rigz1iqh4ymfkgunq6t31u3iycbpr, accessed on 1 June 2020). Data is immediately uploaded to a web-linked *Firebase* console, a Google-owned app development platform (Firebase is accessible here: firebase.google.com, accessed on 22 July 2020). 

Individual playlists were generated in R. These playlists were counterbalanced and involved assigning randomized voice-to-prime type pairings. Each playlist included three voice-to-prime type blocks (e.g., black horse blocked with NZ1, kiwi blocked with NZ2, kangaroo blocked with NZ3). Block order was randomised within and between playlists. Participants only heard each word once, and every playlist incorporated a quasi-random word-to-block order (i.e., no playlist had the same 60 words blocked to a voice compared to another playlist). Participants encountered one block at a time, where each block included 60 unique words with one prime-type before proceeding to the next block. No playlist was ever used twice. The experiment randomly assigned each participant a unique playlist that allocated them to one of four conditions above (see Figure 4a–d). The experiment was designed for cross-participant comparison, so each participant only ever encountered one condition. Participants heard both stimuli and filler words over the course of a session. Blocking voices to prime-type ensured participants would make no crossing associations between a particular prime-type and a particular voice if we were to randomise prime-type and stimuli.

Experiment one briefed participants to listen to English words that may be real or not real and asked them to press key ‘n’ for ‘no’ (not a real word) or key ‘y’ for ‘yes’ (real word). The instructions also suggested that participants use two hands for this experiment, keeping one index finger on key ‘y’ and the other on key ‘n’. Participants were able to take a break between each block and were prompted to rest before they continued. In turn, allowing us to minimize listener fatigue. The full experiment took an average of 20 minutes to complete. A questionnaire followed the listening portion of the experiment. It concerned participant background and attitudes towards Australia and New Zealand. Last, we gave participants a debrief to read and a secondary ‘yes’ or ‘no’ consent option.

#### 3.1.5. Recruitment and Participants

The experiment was available to participants on the internet, and we recruited participants through personal connections and paid Facebook advertising. The advertisement described the experiment as a word recognition task. Participants were told that we were interested in seeing how New Zealanders hear words. It asked that participants wear earphones/headphones when doing the experiment. The advertisement also included an incentive of NZD$10 e-voucher which eligible participants could optionally claim. We limited the experiment to NZE speakers who have no hearing impairments, learned English as one of their first languages, and have lived in New Zealand since the age of seven with no extensive gaps (over 1 year). 

Following our preregistered criteria, we excluded data which was 2.5 sd outside the mean; if a participant answered *yes* to having a hearing impairment; if the participant did not learn English from birth and/or if participants were not living in New Zealand since before the age of seven; or have lived outside New Zealand for more than a year. Furthermore, we excluded unreliable data if the filler words were answered below 68% accuracy (2.5 sd from the mean). After excluding outliers, 119 participants were eligible for analysis (one short of our pre-registered minimum of 120). Their distribution across conditions is shown in Table 4.

### 3.2. Preregistered Predictions

These are the predictions that appeared in our preregistration (we have reworded these slightly to align with the terminology we have adopted in this paper, but we have not changed the predictions. See https://aspredicted.org/rw4kq.pdf (accessed on 18 May 2022)):We expect the kangaroo to increase ‘yes’ responses to KIT-frames and decrease them to DRESS-frames.We expect the differences between the animal primes to be greater in the priming-speaking than in the prime-incidental condition.We do not expect differences between the all-horse baseline presentation-types or the different horses within these conditions.There may be block and trial effects.Predictions (1) and (2) may be mediated by listener gender or Australian English experience or Attitudes toward Australians.

### 3.3. Statistical Approach

The effect of the prime was not retained as significant in our preregistered model: 

(a) Grouping Prime-Speaking and Prime-Incidental Conditions together in one model (with PrimeType having 3 levels—horse, kiwi, kangaroo), and grouping AllHorse-Speaking and AllHorse-Incidental Conditions together in another (with PrimeType having 3 levels—horse1, horse2, horse3) we will test: YES ~ PrimeType × StimulusType × PresentationType + Block × Trial-within-Block + (1 + PrimeType + StimulusType + PresentationType|Speaker) + (1 + PrimeType + StimulusType|Listener) + (1 + PrimeType + PresentationType|Word).

However, while our preregistration indicated a possibility that there would be block effects, our preregistered model did not allow for the possibility that the effect of the animal might differ strongly across blocks. Exploration of the data indicated a strong effect of prime in the first block only, which then seemed to persist through the remaining two blocks. As such, the effect of the animal on the screen in the later blocks appeared counter to hypothesis (e.g., if the kiwi was shown first, the effect of the kiwi was still evident in the data when the kangaroo was later shown), cancelling out any effect in any model that did not take this persistence into account.

In order to assess the significance of these apparent effects, we modified our planned modelling procedure to take into account two non-planned factors—a binary category indicating whether the block is the participant’s first block, and a category indicating which animal was the animal displayed to the participant during the first block.

Two separate generalised linear mixed-effects regression (glmer) models were implemented for the baseline condition and the priming conditions data, initially starting with the same effects and interactions. We used a backwards stepwise procedure involving ANOVA comparisons find the best model per dataset. 

The dependent variable in the model is our estimate of what vowel our participants appeared to hear in the experiment—either a KIT or DRESS vowel (we note this dependent variable also departs from our preregistration, which planned to model whether the participant answered). This estimate was coded as KIT-consistent if participants answered ‘yes’ to a vowel in a KIT frame (such as dXgging), or ‘no’ to a DRESS frame (such as sXnding). The opposite set of answers (no to a word like dXgging and yes to a word like sXnding) were coded as DRESS-consistent. It should be noted that this process involves some assumptions and likely over-simplifications. When someone answers ‘no’ to dXgging, they may have heard it as ‘degging’, but it is also possible that they heard an alternative vowel, such as ‘deeging’. The results from Shaw et al. (2018) [31] (outlined above), show that the most common mishearing of an Australian KIT vowel is DRESS, and vice-versa, but it is not the only mishearing. We also conducted preliminary exploratory modelling that treats the DRESS-frame and KIT-frame words separately and found the same key results as reported in this paper. Grouping the results together into the same model leads to greater clarity and fewer models. In all cases *frame* × *prime* is tested, to allow for the possibility that the responses to the two frame types should be treated separately.

Two sets of interactions were tested in the modeling procedure. The first set involved the prime present on the screen. However, given that experiment one was a within-participant blocked design, and following exploratory analysis, we also wanted to allow for the possibility that the prime presented in the very first block would have a pervasive effect. We therefore also included a set of interactions involving the identity of the first prime. 

We fit down from:

KIT-consistent-response ~ primetime × presentationtype × (firstblock + frametype + orderwithinblock) + firstprime × presentationtype × (firstblock + frametype + orderwithinblock) 

Random effects of *id* (individual participants) and *word* (stimuli) were included. Slopes were explored but led to non-convergent models, and so were dropped following the preregistered procedure, which was retained for the baseline model, and retained through most of the modeling for the priming model, but then dropped to obtain convergence. Early modeling included speaker intercepts representing the 3 different speakers, but these led to multiple convergence issues and explained little variance, so they were dropped (dropping slopes to fix convergence problems was anticipated in the preregistration).

The fitting procedure involved iteratively removing first interactions then main effects and comparing minimally different models via ANOVA comparisons. If an interaction or main effect did not lead to a significantly improved model, it was excluded. 

### 3.4. Results

The baseline model revealed no significant interactions, and two main effects. The main effects were first block—with responses in the first block leading to more KIT-consistent responses, and frame-type, with KIT frames eliciting more KIT-consistent responses. Importantly, which of the three horses was presented had no significant effect, consistent with prediction (3).

The prime model is shown in Table 5. Like the baseline model, it also includes the increased KIT-consistent response in the first block, and a bias toward more KIT-consistent responses for KIT-frames. It contains an additional order effect, in which the proportion of KIT responses reduces slightly over the course of each block. Presentation-type is not retained as significant. The non-significance of presentation-type contradicts prediction (2) that a prime presented ‘speaking’ the stimuli should invoke greater perceptual shift in listeners. 

Counter to prediction (1), there is no significant effect of the prime-type. However, while the on-screen prime had no significant effect upon participant responses, we did find a significant interaction between the prime presented in the first block and the frame-type. The first prime listeners encountered in the experiment shifted their perception boundary for DRESS words and this persisted in their responses for the remaining two blocks. Once listeners began hearing the stimuli a certain way, it remained this way for the rest of the experiment. 

This interaction is shown in Figure 5. With ‘percentage KIT-consistent responses’ on the y-axis, if our prediction that the kangaroo prime will induce greater KIT responses (cf. prediction (1)) is borne out in the data, then the blue kangaroo point would be the highest out of three prime points on the y-axis. Likewise, we expect the green kiwi point to be the lowest out of the three prime points because we predicted that the kiwi prime would induce fewer KIT responses (and inversely, greater DRESS responses). The graph is divided by DRESS frames and KIT frames on the x-axis to show the significant interaction. Looking first at the DRESS frame, Figure 5 suggests that the kiwi prime shifted listener perception in the predicted direction. In other words, listeners who saw the kiwi prime in the first block were more likely to give DRESS-consistent responses. 

We note that the horse is not positioned between the kangaroo and the kiwi. Rather, the horse elicits the most KIT-consistent responses, and is not significantly different from the kangaroo. The kiwi elicits the fewest KIT-consistent responses, and relevelling the model confirms that it is significantly different both from the kangaroo and from the horse baseline (with kiwi as intercept: horse est = 0.84, *p* < 0.001; kanga est = 0.57, *p* < 0.05; KIT × horse est = −0.89, *p* < 0.0001; KIT × kanga est = −0.82, *p* < 0.001). Thus, while the difference between the kangaroo and the kiwi is as predicted, the inclusion of the horse reveals that the effect is being driven by the kiwi. 

Turning now to the KIT frame, Figure 5 shows a likely ceiling effect for all three primes. This effect suggests that when presented in a KIT frame, listeners hear the ambiguous stimuli as KIT regardless of what prime they saw in the first block.

### 3.5. Interim Summary

This experiment supported our overall hypothesis that participants would report hearing more KIT-consistent vowels when primed with a kangaroo than a kiwi. However, it had several limitations. First, the within-participant design did not work. Once exposed to a prime, participants maintained their behaviour throughout the rest of the experiment. Second, the stimulus was clearly not ambiguous enough. Responses were overwhelmingly associated with ‘KIT’, even when for the DRESS-frames which are expected to induce a perception of DRESS due to the Ganong effect [36]. Likewise, the KIT frames were responded to positively at a rate close to ceiling. A third limitation is that the significant difference between the kangaroo and the kiwi is unlikely to be caused by a strong priming effect of the kangaroo (as predicted). Rather, the greatest departure from the baseline horse responses is observed in the kiwi condition; the difference between the horse and the kangaroo is not significant and is in the opposite direction as predicted. One interpretation of the greater effect of the kiwi prime may be that listeners did not begin the experiment in a ‘New Zealand English’ listening mode. This response could be caused by the online environment, in which many accents are heard, together with the acoustic properties of the target vowel which (by design) did not actually match either NZE or AusE completely. Thus, the default in this context may have been to expect an ‘other’ accent, and it is only the presentation of the kiwi that triggers a more NZ-like listening model and accepts the ambiguous vowel as a viable ‘DRESS’. We will return to this suggestion in the discussion.

While this experiment provides some overall support for the hypothesis that kangaroos and kiwis can elicit different speech perception behaviors, it also has the above limitations. We attempt to rectify these issues and replicate the key result in experiment two.

## 4. Experiment Two

### 4.1. Methods and Materials

A modified second experiment was conducted as a follow up to the first experiment. In particular, given that the blocked design was not successful, we switched to a cross-participant design in which each participant was only exposed to one prime. The experiment remained a lexical decision task with some changes explained below.

#### 4.1.1. Conditions and Prime Types

We changed the experiment to be a cross-participant design, where listeners would only see one prime-type. Furthermore, we excluded the baseline condition (three horse primes) given that we know the condition is performing as expected and is having no influence on listener perception. We maintained the speaking/incidental condition to ensure its influence, or lack thereof, in listener perception. Participants thus saw one of six images (a horse, kiwi, or kangaroo, in either speaking or incidental condition). Participants all responded to the same stimuli, in the same voice, but in different random orders.

#### 4.1.2. Stimuli

Given the apparent ceiling effect of the perceived KIT vowel, we decided to use stimuli that were at step 4 of the vowel resynthesis (refer to Figure 2 to see synthesis step 4 in the continuum). This stimulus is a step closer to an Australian KIT (and thus New Zealand DRESS) vowel; it is more likely to be heard as DRESS by our NZE participants and should therefore reduce the KIT ceiling effect. Speaker NZ1 was the voice used for all stimuli. The total number of word-types were dropped from 180 in experiment one to 160 in experiment two (80 target stimuli and 80 filler stimuli). 

#### 4.1.3. Online Word Recognition Task

Experiment two followed the same procedure as experiment one: that is, using the *Speech In Noise 2* program to run the experiment online, collating data to a new Firebase console. Like experiment one, participants were asked to listen to English words which may be real or not real and asked them to press key ‘n’ for ‘no’ (not real word) or key ‘y’ for ‘yes’ (real word). Following the listening task, we added six post-listening questions regarding the prime pictures—“What animal is this?” (where participants would type an answer), and “Does this animal suggest any particular country to you?” (where participants could select one from multiple answers). The addition of these questions reinforced that the prime animals were being accurately identified, and associated with the target country (e.g., the kangaroo prime could potentially be confused as a wallaby, which lives in both Australia and New Zealand). The results of this task showed high identifiability of the animals, and reliable associations with the target country for the kiwi and the kangaroo. The same personal and attitudinal/exposure questionnaire used in experiment one was used to finish this experiment. 

#### 4.1.4. Recruitment and Participants

We used the same recruitment, payment and inclusion criteria described in experiment one. After filtering out responses from participants who did not meet the criteria, and those whose data were 2.5 sd outside the mean, data from 136 participants were eligible for analysis. The distribution of speakers over conditions can be seen in Table 6.

### 4.2. Statistical Approach

We did not separately preregister experiment two. However, in our modeling, we undertook a two-step procedure that followed the spirit of our original preregistration. In a first step, we modeled overall effects, without regard to social factors, following (the spirit of) the model in our original preregistration. In a second step, we attempted to investigate any mediating effects of social factors. The modeling procedure for this step is described in detail in the Appendix A (available here: https://github.com/jenniferhay/kangaroo-kiwi (accessed on 1 June 2020)). The data contained significant four-way interactions, which ultimately led us to a modeling procedure which dealt with the men and the women in separate models. We considered effects of *prime-type*, *order* and *presentation-type*, in addition to social characteristics of *age*, *gender*, *experience* (amount of time spent in Australia), and *attitude* (numerical scale based on post-questionnaire attitude questions). Random effects of *id* (individual participants) and *word* (stimuli) were included. Like experiment one, model slopes lead to convergence issues and are not pursued. 

As described below, both the men’s and women’s models included effects of prime-type and order within the experiment. The women’s data also included significant interactions involving social characteristics.

### 4.3. Results

#### 4.3.1. Primary Results (No Social Factors)

We did not update our preregistration between experiments one and two, but in this analysis, we largely followed the model preregistered for experiment one.

The preregistered fixed effects for the experiment one model were:

primetype × frametype × presentationtype + block × trial-within-block.

Experiment two did not contain blocks, so block was not included in the experiment two model. Informed by the results of experiment one, we also wanted to allow for the priming effect to evolve over the course of the experiment. To this end, we included trial order within the interaction rather than as a separate effect, testing a four-way interaction between *presentation-type*, *order*, *frame-type and prime-type*. We simplified the random effects to obtain convergence, following the procedure outlined in the preregistration. The effect of order was scaled and centred.

Pruning this model, we observed a significant effect of prime-type, in interaction with frame-type and order (shown in Table 7). Interactions involving presentation-type resulted in significant improvement to the model, but these models failed to converge. Checking separate models of the different frame-types yielded convergent models with no significant effect of presentation-type. Presentation-type was then dropped from the model.

The significant three-way interaction can be seen in Figure 6. Contrary to the result from experiment one, the primary difference between the kangaroo prime and the kiwi prime is now found in the KIT-frame rather than the DRESS-frame. In the DRESS-frame, while there appear to be some differences involving the horse, the kangaroo and kiwi are not different from each other. In the KIT-frame, however, we see a general decline through the experiment in the KIT-consistent responses—except in the case of the kangaroo, which maintains a consistently high rate of KIT-consistent responses.

Thus, this model provides further support for predictions (1) (the effect of the animal prime on perception), and (3) (the possibility of order effects). It does not, however, support prediction (2) (the stronger effect in the speaking rather than incidental conditions).

#### 4.3.2. Social Factors

Our preregistration also flagged social factors as an interest, and prediction (4) noted that we expect certain social factors may be playing an important role in this work. We explore the roles of these factors in the analysis below. The models we pursued involved fitting multiple subparts of the data to reveal sub-regularities without striking convergence issues. While the general social factors explored echoed those anticipated in the preregistration, the specific models were more complex; the preregistration planned simple interactions between individual social factors and the prime, without consideration of a mediating effect of order or the interactions between social factors themselves. Because we deviated significantly from the planned structure, and found some unpredicted effects for social characteristics, the model fitting reported in this section should be regarded as strictly exploratory.

Our attempt to explore social factors revealed some differences in the men versus the women in the experiments (two nonbinary participants were omitted from this secondary analysis—see models and model fitting procedure in Appendix A), resulting in separate models for the men and women. Due to gender imbalance in the participants, the model for the women contains more than twice as many participants as that for the men. 

In addition to the overall KIT-frame result found in Section 4.3.1, the men also showed an effect in the DRESS-frame that emerged over the course of the experiment. In this model, we observed increased KIT-consistent responses to the kangaroo and decreasing KIT-consistent responses to the kiwi (Figure 7—consistent with prediction (1)).

For women, the priming effect appears to be mediated by time spent in Australia. Listeners who have spent more than a month in Australia show the predicted effects for KIT and DRESS. Listeners who have spent less than a month in Australia, however, show no effect for KIT and an effect in the non-predicted direction for DRESS (Figure 8). A shift in the non-hypothesized direction by any group of participants was not predicted, but it is consistent with some results in the wider literature that will be discussed below.

Prediction (4) suggested that any observed priming effects may be “mediated by listener gender or AusE experience or Attitudes towards Australians”. Exploration of these social factors supports that these effects do not behave the same way for all participants, consistent with earlier work (cf. Hay, Drager and Nolan 2006; [1]). However, as these results reflect modeling that incorporates a number of social factors, which carved data up in a number of ways, they should be taken as tentative. It is certainly not guaranteed that another sample would yield these same interactions. What the interactions suggest, however, is that different participants or groups of participants respond to the stimulus and primes somewhat differently. It is likely that the locus and even direction of the effect may be mediated by the relationship between the stimulus acoustics, the participant’s own production, and the participant’s previous experiences and stereotypes.

## 5. Summary

Experiment one showed a strong priming effect, in which the kiwi (relative to the kangaroo) elicited fewer KIT-consistent responses (i.e., answered ‘yes’ more) to vowels in a DRESS-frame. This effect persisted through blocks and was not overridden by subsequent presentations of other animals. In this experiment, the kangaroo appeared to act more similar to the horse, and the kiwi looked to be the image that produced different behaviors in the participants.

Experiment two changed to a cross-participant design, with a different stimulus that aimed for a greater level of ambiguity between DRESS and KIT. In this experiment, our overall analysis found an effect in the KIT-frame where the kiwi aligned with the horse and the kangaroo elicited a different response, emerging over the course of the experiment.

The exploration of social factors in experiment two also indicated some differences across participant subgroups. Men showed priming in the predicted direction for both frame-types in an effect that emerged over the course of the experiment. Women with some exposure to AusE showed the expected effect for both KIT and DRESS frames. However, women with very little exposure showed no effect for KIT frames and an effect in the wrong direction for DRESS frames.

In terms of the preregistered predictions, we can summarize the results as follows.

We expect the kangaroo to increase ‘yes’ responses to KIT-frames and decrease them to DRESS-frames.

Overall, this prediction was supported. In both experiments, we did find a difference between the animals in the predicted overall direction, with the kangaroo eliciting more KIT-consistent responses than the kiwi. In experiment one, the initial presentation of the kiwi (relative to the kangaroo AND the horse) elicited more ‘yes’ responses to the DRESS-frame in an effect that persisted across blocks. In experiment two, the kangaroo elicited more ‘yes’ responses (relative to the kiwi and the horse) in the KIT-frame, a result that emerged over the course of the experiment.

2.We expect the differences between the animal primes to be greater in the priming-speaking than in the prime-incidental condition.

We found no evidence in support of this prediction.
3.We do not expect differences between the all-horse baseline presentation-types, or the different horses within these conditions.

This prediction was supported.
4.There may be block and trial effects.

This prediction was supported. The block and trial effects were significant, and incorporating them into our analysis required deviation from our preregistered modeling procedure.

5.Predictions (1) and (2) may be mediated by listener gender or Australian English experience or attitudes toward Australians.

This prediction was tentatively supported. Separate models fit to data from male vs. female participants show different results. The men’s data were consistent with our overall prediction (1), whereas the women’s data were mediated by experience with Australia. Results from this social analysis should be regarded as tentative; however, because they were found through extensive post hoc exploration of the data.

## 6. Discussion

When we consider our two experiments together, we find good evidence that priming with regionally-associated animal images can affect patterns of speech perception. However, the location and nature of such effects seem to rely on alignment between participants’ own productions, the nature of the acoustic stimuli, as well as experimental design. In this section, we explore several issues arising from the results reported above.

One thing to consider here is the social and physical differences between our primetype cartoon drawings and that of physical stuffed toys used in Hay & Drager (2010) [1]. The design of our experiment meant listeners were participating virtually while looking at a virtual animal, unlike Hay & Drager (2010) [1] who had in-person participants and handleable stuffed toys. These physical toys could potentially have carried greater social weight for the participants, particularly because these types of toys are often bought as souvenirs. Therefore, it would not be hard to imagine that participants realise some sort of social significance towards the toys and any inherent history that may belong to them. Our line-drawing prime-types are not necessarily associated with exactly the same social meaning. However, despite this, we still see some evidence of social priming with the online line drawings.

### 6.1. Lack of an Effect of Incidental vs. Speaking

A number of different projects have reported effects of expectations or beliefs about the speaker. If a listener is led to believe that a speaker is male vs. female [3], Canadian vs. American [9], older vs. younger [6], or working-class vs. middle class [5], these beliefs can affect their reported speech perception patterns. Munson et al. (2017) [37] show that the effect of gendered expectations on the perception of fricatives is weaker when the gendered expectation is implied (via what was said), as opposed to explicit (via photos).

The specific pair of papers we set out to pseudo-replicate found similar patterns of priming by writing ‘Australian’ vs. ‘New Zealander’ on the response form [10], and by priming with stuffed toy animals in the room [1]. Even though the strength of the effects in Hay, Nolan, Drager (2006) [10] and Hay & Drager (2010) [1] did not appear markedly different, we nonetheless expected that if the animal was represented as speaking, it would have a stronger effect than if it was incidentally presented on the screen. The intuition behind this prediction was linked to the fact that there is an extensive literature (cited above) showing that beliefs about a speaker can influence speech perception, whereas there are few studies that have looked at incidental social priming. While a greater number of studies need not necessarily imply a stronger effect, Munson et al. (2017) [37] finding that somewhat more explicit priming of gender reveals a stronger effect is consistent with our prediction.

This is the first study that used both types of priming in the same task to overtly test whether the effects are of the same type and magnitude. With regard to speech processing, the literature holds more work explicitly exploring the role of the speaker than studies about incidental priming patterns, or the differential effects reported by Munson and colleagues. Considering the available information, we predicted stronger effects might be observed when our animals appeared to be talking; this prediction was preregistered (as prediction (2)). Indeed, notwithstanding the literature, we would also presume that there is a significantly increased utility of adjusting one’s speech perception strategy in response to the speaker, as opposed to responding to seemingly unrelated images or objects. This informal assessment gave rise to the expectation that we would see a difference between the speaking and the incidental conditions.

However, across both experiments our final models did not retain presentation-type as a viable predictor. The non-significance of presentation-type reinforces the interpretation that what we are seeing is a relatively automatic priming effect in which animal images are sufficient to activate the social category of ‘Australian’ or ‘NZ’. This activation, in turn, likely progresses jointly with speech memories and/or models that are indexed to these social categories.

Of course, there are limits to our ‘speaking’ condition that must be considered in interpreting this result. While film, media and television have acclimatized us all to the idea that cartoon animals come from particular places and have particular accents, it is also true that most of us know that the voices produced by cartoon animals are actually voiced by a third-party voice-actor. That is, no one really believes that these animals are actually producing the speech that we hear. Even in our ‘speaking’ condition, then, our participants would not truly believe that they are listening to a voice actually produced by the animal. In that sense, perhaps the use of animal images makes both of our conditions somewhat ‘incidental’—that is, the listener knows (at least at a subconscious level) that the voice has been produced by someone not seen on the screen. For a truer ‘speaker’ condition, we would need photos of actual people, and to lead the listener to believe the voice is truly produced by those people.

We thus have good evidence for the contribution of incidental priming, and that it does not matter whether an animal is represented as speaking or not. A stronger comparison of incidental vs. speaking effects would require an altered methodology, which must include “speakers” that are not so easily ruled out or dismissed.

### 6.2. The Acoustics of the Stimulus and the Locus of the Effect

There is a clear Ganong effect in the results, wherein listeners are more likely to interpret our ambiguous vowel as a phoneme that would complete a real word [36]. The average by-item ‘yes’ response for words containing the ambiguous vowel was 71%. Some word frames elicited this effect much more strongly than others. The frame most often identified as a real word was ‘tXxture’ (97%), whereas the least was ‘thXnnest’ (17%). A large amount of variation is no doubt linked to coarticulatory effects, word frequency, and the distinctiveness of the phonological frame. Some are also linked to specifics of the wider frame. We noted, for example, low rates of ‘yes’ responses to some filler items that should have been heard as words for both DRESS and KIT, such as ‘bXdder’. Double-checking of these recordings revealed that the second syllable was produced with a degree of stress and a NURSE vowel rather than a schwa vowel, which would be more common in NZE, likely leading listeners to hear it as a nonce compound. We did not exclude any items from our analysis but rather relied on the random intercept for word to have controlled for this variation. It is very possible that priming occurred to greater or lesser degrees in some words depending on the strength of bias of other factors.

One respect in which our experiments differ from Hay, Nolan & Drager (2006) [10] and Hay & Drager (2010) [1] is that we have included a baseline condition. Adding this condition enables us to tell not only whether there is a difference between the kangaroo and kiwi conditions, but also whether one or both differ from a baseline in which there is no strong regional prime. While we found differences between the kangaroo and the kiwi in both of our experiments, the relationship with the baseline differed. In experiment one, the kangaroo and the horse elicited similar responses, and the kiwi elicited more DRESS-like responses. The direction of this prime is in the predicted direction, but the fact that it is the kangaroo (rather than the kiwi) that patterns with the baseline was not predicted.

Our interpretation of this result is that listeners may not have initially engaged with the task expecting that they were listening to a New Zealander. The task was conducted online, where the majority of voices one hears are probably not NZE voices. The voice encountered in our stimuli presents an ambiguous vowel which is not exactly typical of NZE or AusE, but instead exists in a perceptual space utilized by some more remote dialects (cf. [31] data, above). In this interpretation, listeners do not automatically/necessarily engage with the voice as if it is a New Zealander, except in the case where they are presented with a kiwi prime. The kiwi prime elicits a more NZE like listening mode and, thus, more ‘DRESS’ responses. This interpretation is also consistent with the high KIT ceiling-effect observed in experiment one: many dialects produce KIT somewhere in the range between the NZE and the AusE variant, but none of the other dialects in Figure 1 contain a DRESS vowel in that vicinity. The consequence with respect to priming is that a KIT-consistent response appears to be the default. Priming mainly occurs when the DRESS-frame and the kiwi prime are combined, two factors that together can increase a DRESS-consistent response.

Experiment two shifted the ambiguous stimulus to be higher and fronter in the vowel space. The stimulus is now not so close to a NZE KIT vowel, but close to an AusE KIT vowel and a NZE DRESS. This change had the overall effect of slightly increasing DRESS-consistent responses. Overall, the responses shifted from 57.5% KIT consistent in experiment one (78.6 for KIT frames and 36.5% for DRESS frames) to 50.4% KIT consistent in experiment two (73% to KIT frames and 27.9% for DRESS frames). In the main analysis of experiment 2 we see the locus of priming occurring when we have a combination of the KIT-frame and the kangaroo. It seems likely, then, that shifting toward a stimulus that is closer to something a New Zealander produced increased the degree to which listeners engaged with the speaker as a New Zealander; the priming effect which shifts perception away from this mapping is driven by the kangaroo. In each experiment, the prime that differs from the horse is the one that is aligned with the frame (kiwi for DRESS-frame and kangaroo for KIT-frame). A categorization that is consistent with the frame (i.e., resulting in a word) is further facilitated by priming with regional primes.

### 6.3. Effects of Blocks and Order

In experiment one we attempted a within-participant design, expecting that participants’ responses to our stimuli would change when we altered the visual prime. Across three blocks we changed the voice of the speaker, and the image that was being looked at. However, while the perceptual behavior exhibited in the first block showed evidence of priming across participants, this influence upon perception appeared to persist for participants across subsequent conditions for the remainder of the experiment.

While we used three different voices in experiment one, we would like to draw attention to the fact that these voices were not radically different from one another. They were all synthesized combinations of the same Australian female, together with each of three young New Zealand female speakers. While our impression was that the speakers were distinguishable, the speaker change across blocks may not have been particularly apparent to our listeners, who were simply working quickly and focused on finishing the task at hand. Switching to radically different voices between blocks may have better facilitated a ‘reset’ of the listening behavior for participants, and thus allowed for increased influence through a new prime. We included some informational screens between blocks, but the pause between blocks will have been minimal for most participants. A between-blocks distractor task that was completely unrelated to the experiment may also have facilitated a more successful within-participant design.

This type of effect, involving initial exposure conditions persisting through later conditions, have been well-documented. Bordens and Abbot (2002) [38] describe such carryover effects as one of the most serious disadvantages of a within-participants design. Recent examples involving speech perception include Hay et al. (2017) [30], who examined whether the vowel boundary on a DRESS-TRAP continuum was affected by the location of the listener. The authors found a significant difference between the locations of the first completion of the task, but this perception persisted to the second location. Hashimoto (2019) [39] looks at social (topic-based) priming in a speech production experiment and shows persistence effects across blocks.

When we shift to an across participant design in experiment two, we see an effect emerging over the course of the experiment. A general trend towards decreasing KIT responses is absent when the kangaroo is the prime-type, leading to a significant separation at the end of the experiment between the kangaroo condition and the kiwi/horse conditions. This order effect may seem to contrast with experiment one in which we reported a persistent effect, with no effect of order. However, in posthoc analysis, we can see a trend toward a non-significant order effect in experiment one as well. The difference between the two experiments is likely the length of the analysis period and the exposure—just 30 target items in block one of experiment one, vs. 80 for experiment two, lending more granularity to an analysis of order. 

### 6.4. The Role of Experience versus Stereotypes

While the priming generally shifted responses in the predicted direction, there is one pocket of data where effects were observed in an unanticipated direction. Women with little prior experience of AusE (i.e., have spent less than a month in Australia), showed no priming of KIT word-types, and an effect in the unpredicted direction for DRESS word-types. That is, when the frame was DRESS, they were more likely to answer ‘yes’ when looking at the kangaroo than the kiwi or the horse. While this result was not predicted and should be regarded as tentative, it is nonetheless consistent with some previous results in the literature that seem to reflect some New Zealanders holding a false stereotype of a high Australian DRESS vowel.

Ludwig (2007) [40] shows that, when presented with words in isolation and asked to rate them as produced by a New Zealand or Australian speaker, New Zealanders are very good at rating them when it comes to KIT but not when it comes to DRESS. In her study, Ludwig suggests that New Zealanders are prepared to accept both the New Zealand and the Australian variant as a NZ DRESS, unless it was nasalized, in which both nasalized variants were perceived as Australian. This pocket of the data, then, connects to a thread of literature suggesting that primes can sometimes activate stereotypes in the absence of extensive experience with a dialect.

False stereotypes about AusE DRESS are likely to be less common among individuals who have more direct experience with AusE. There is some evidence that differences in experience influence behavior in production tasks. Sanchez, Hay & Nilson (2014) [41] explored the effect of conceptual activation of Australia on New Zealanders’ vowel productions, using both corpus analysis and a word-list elicitation task. In the corpus analysis, they found that talk about Australia shifted KIT and TRAP in a more Australian direction. Both the corpus and the wordlist reading showed an unexpected interaction for DRESS. Speakers who have experience with AusE produced more Australian-like DRESS in contexts that were primed with Australia. However, speakers with less Australian experience actually produced less Australian-like vowels in the Australian-like context. Sanchez, Hay & Nilson (2014) [41] argue that the non-experienced New Zealanders are influenced by a false stereotype that Australians produce high DRESS vowels.

We have three possible explanations for why this effect was seen within the women but not the men. One is simply that we do not have many men in our study and might observe a more congruent effect through further sampling. A second explanation is that there may be some genuine difference in stereotypes, experience or attitudes that align with gender. This explanation would be consistent with the gender differences also observed in Hay, Nolan and Drager (2006) [10] and Hay & Drager (2010) [1], and the exploration of attitude by Walker et al. (2018) [24]. Finally, a third possibility is that there may be a difference in the participants’ own production that leads to different behaviors in the task. There is certainly evidence that individual variation in production of various sounds can be related to their perception [42,43,44]. We have seen between experiment one and experiment two that a subtle shift in the acoustics of the stimulus matters. It seems possible, then, that subtle differences in the participants’ own production could influence their perceptual processes. Based on the literature exploring DRESS raising within New Zealand, we might expect the men to have lower DRESS vowels than the women (e.g., [45]), thus making the stimulus a potentially close match to their own DRESS vowel than the women. 

Indeed, for the participants in the experiment 2 ‘neutral’ horse condition, the men responded with 41% KIT-consistent responses as opposed to the women with 51%. It is thus likely the men identified the speaker as a New Zealander, rather than an overseas accent with a non–local KIT vowel, causing the baseline and the kiwi to group together. This slightly greater bias toward hearing DRESS may have increased the potential for priming towards KIT with the kangaroo, even in the DRESS frame. In sum, when there is already a KIT bias (experiment one) the kiwi can push the listener more towards hearing DRESS. When there is a DRESS bias (experiment two, for men) the kangaroo can push the listener more towards hearing KIT. In the most balanced of our subsets (women in experiment two), we see some evidence of priming by both animals, however, when the women have little prior experience of Australia this effect appears to be influenced by stereotype rather than experience.

Drilling too far into the shifting loci of the effect would drift into conjecture, though it becomes clear as we shift between different acoustic stimuli and different subpopulations who themselves no doubt have different productions (men vs. women), stereotypes and experiences, that the ability to manipulate perceptions in different directions is inconsistent. The distance of the stimuli from a local production may also lead to differing assumptions about where a speaker is from, which can also influence the baseline in different ways. A study which explores individual participant’s productions, and the relevant vowels’ distances from our stimuli might help shed light on the dynamics at play here.

### 6.5. Limitations

In the present work we set out to address the limitations in Hay, Nolan & Drager (2006) [10] and Hay & Drager (2010) [1]. We addressed these limitations by designing a new task that is more clearly linked to speech perception, by incorporating larger participant samples, by preregistering our predictions, and by repeating the task across two different versions of the experiment. In doing so, we found evidence to support the claim that priming with regionally-associated animals can affect speech perception patterns. 

However, we departed from our preregistered model in various ways, and in both experiments our reported results involved aspects of exploratory analysis. In experiment one, we needed to isolate the prime presented in block one to explain response patterns later in the experiment. In experiment two, our analysis took into account interactions involving order, which weren’t anticipated in the preregistration. In our analysis of social factors (experiment two), we followed a complex model fitting procedure to find models that both converged and represented the patterns we find in the data (see Appendix A, available here: https://github.com/jenniferhay/kangaroo-kiwi (accessed on 1 June 2020)). For strict proponents of preregistration who are not supportive of exploratory modeling, our results will leave something to be desired.

An ongoing challenge for replication is that these types of results are population-specific, including the particulars of participants’ own production patterns and previous exposure to AusE. Even if we were to repeat the experiment in Christchurch, sound change involving the DRESS and KIT vowels might lead listeners to have a different relationship with our stimuli. Indeed, there is also an ongoing regional variation with DRESS [46] that may also lead to variable behavior in the experiment, and which we have not accounted for here.

Embedded within our results are many reasons why priming effects may be hard to find, and why it might be hard to see them extended across a range of contexts. These effects can be dependent on the specific stimulus and the individual and are sensitive to the experimental design. However, with that said, the above results support claims that such priming effects do in fact exist. We hope that further work can build upon this experimental methodology to explore relationships between speaker production, speech perception, and social primes. 

## 7. Conclusions

To address a variety of limitations noted in Hay & Drager (2010) [1], we employed an alternate experimental design aiming to replicate their main reported effect. We wanted to shift to a design in which the results could more unambiguously be interpreted as reflecting priming-based differences in speech perception.

The design detailed above shows promise as a means to investigate how contextual factors can influence speech perception. Placing an ambiguous vowel in a Ganong context facilitated the perception of that vowel in a word-consistent manner. A categorization that was consistent with the frame (i.e., resulting in a word) was facilitated by priming with the appropriate regional prime. The overall result across two experiments was that exposure to a kiwi facilitates more DRESS-consistent responses to our stimulus, as compared to a kangaroo which facilitated more KIT-consistent responses.

Exploratory analysis of social factors indicated variability across different groups of listeners. One unexpected example that we noted was that women listeners with little previous experience of Australia tended to show an opposite effect for DRESS. This finding is consistent with reports in the literature that New Zealanders with little exposure to AusE may be influenced by false stereotypes regarding their DRESS vowel.

Overall, our attempt to replicate kangaroos/kiwis as affecting speech perception for New Zealanders was a qualified success. On the one hand, we preregistered a new design and found significant results across two experiments consistent with this interpretation. On the other hand, the modeling at every step involved some aspect that was not fully anticipated in the data, and both experiments contain a subgroup within the data where the priming did *not* occur. The success of the research program, therefore, comes with caveats and caution. 

Our own overall interpretation is that these experiments reinforce the idea that such priming effects exist, but also very clearly illustrate the degree to which these effects are mediated by many factors. Such factors are likely to include the experimental design and order effects, the lexical frame, the acoustics of the stimulus, as well as the listener’s own production, previous experience, and stereotypes. We would therefore not anticipate that this exact design would necessarily elicit results in Australia, for example, where listeners have different associations with the animals, and experience with the accents. It would not even necessarily elicit the same results if rerun in New Zealand, given the ongoing changes in KIT and DRESS both in NZE and AusE. Exploring the specifics of the boundaries and limitations on this type of priming will provide an interesting avenue for further research.

## Figures and Tables

**Figure 1 brainsci-12-00684-f001:**
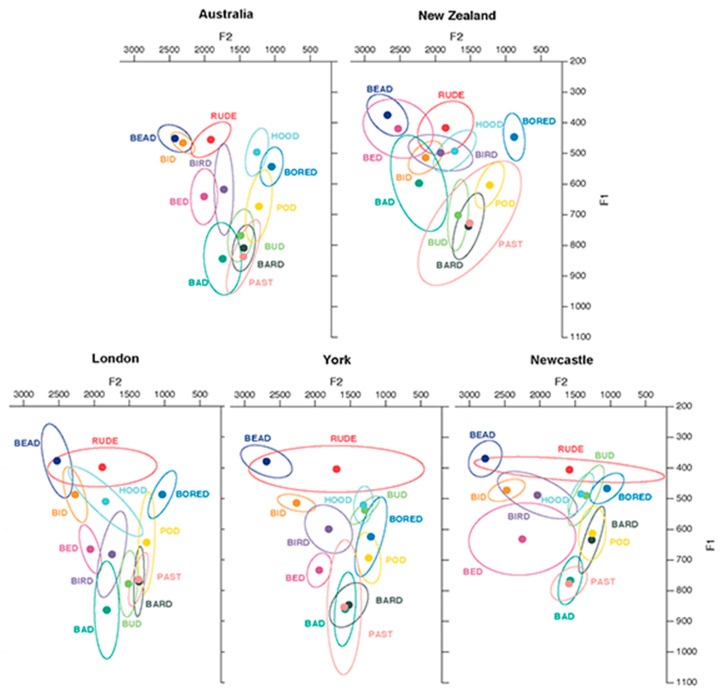
F1-F2 plot of vowels in Australian English (top left), NZE (top right), and three UK dialects (bottom) as reported in Shaw et al. (2018). F1 represents tongue height (high vs. low) while F2 represents tongue backness (front vs. back). For example, *BEAD* in the vowel plots depicts the tongue in the highest and furthest forward position in the mouth. (CC BY-4.0 https://creativecommons.org/licenses/by/4.0/, accessed on 12 February 2022).

**Figure 2 brainsci-12-00684-f002:**
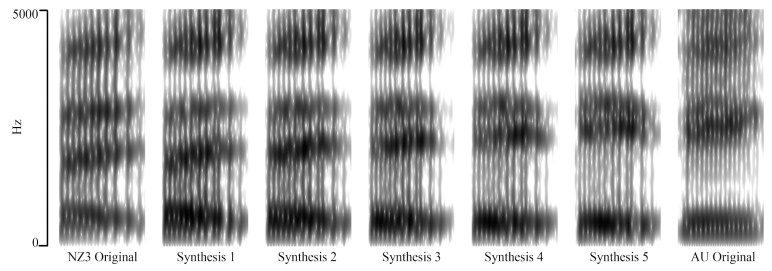
Five-step vowel continuum synthesis of the word ‘fixture’ proceeding from most New Zealand-like (NZ3 original recording) to most Australian-like (AU original recording).

**Figure 3 brainsci-12-00684-f003:**
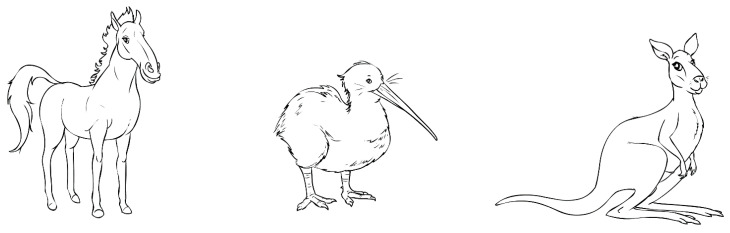
Commissioned artwork of prime-types (horse, kiwi, and kangaroo).

**Figure 4 brainsci-12-00684-f004:**
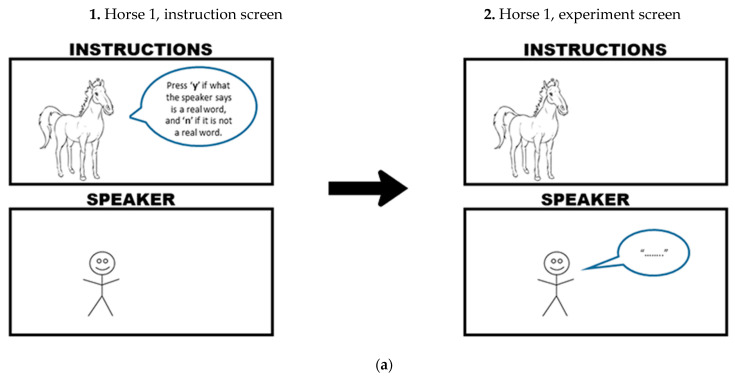
(**a**) Example block from **baseline-incidental** condition. (**b**) Example block from **priming-incidental** condition. (**c**) Example block from **baseline-speaking** condition. (**d**) Example block from **priming-speaking** condition.

**Figure 5 brainsci-12-00684-f005:**
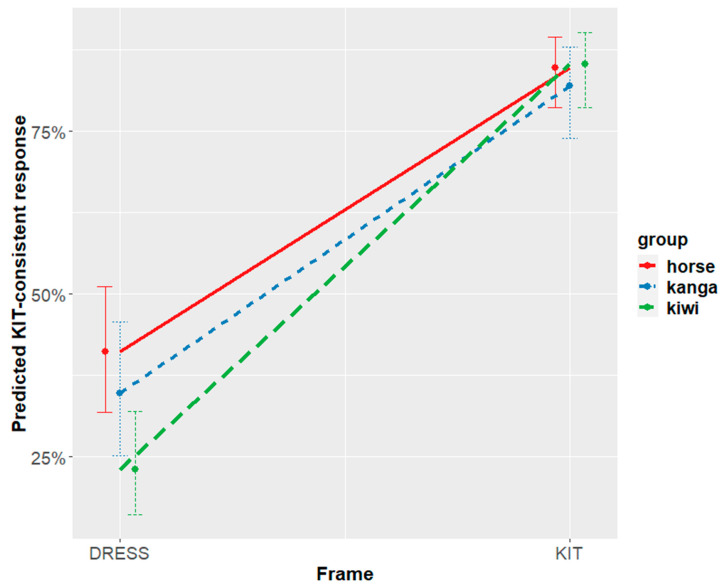
Model prediction plot from the model of experiment one (see Table 5).

**Figure 6 brainsci-12-00684-f006:**
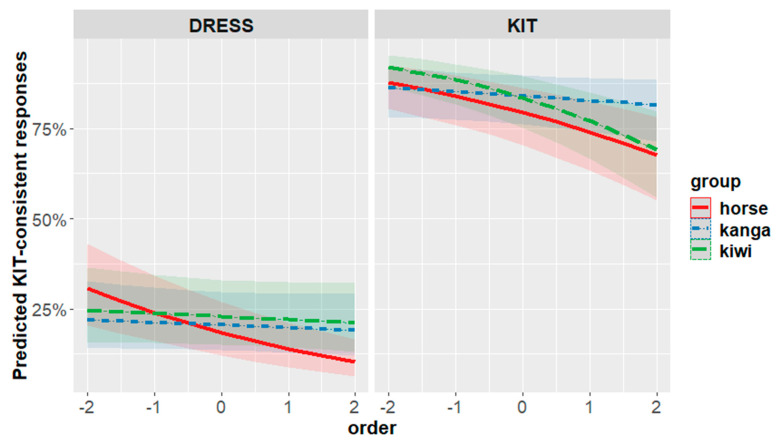
Model prediction plot from model of experiment two (see Table 7). Three-way interaction of order (x-axis), primetime (lines), and frame (panels), predicting percentage of KIT-consistent responses.

**Figure 7 brainsci-12-00684-f007:**
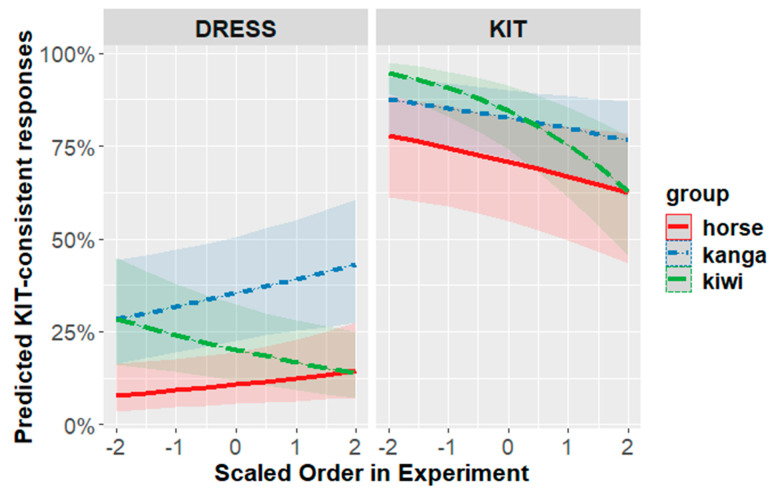
Model prediction plot from the model fit to men only (model details in Appendix A). The figure plots a set of 3 two-way interactions involving order (x-axis), prime-type (lines), and frame (panels).

**Figure 8 brainsci-12-00684-f008:**
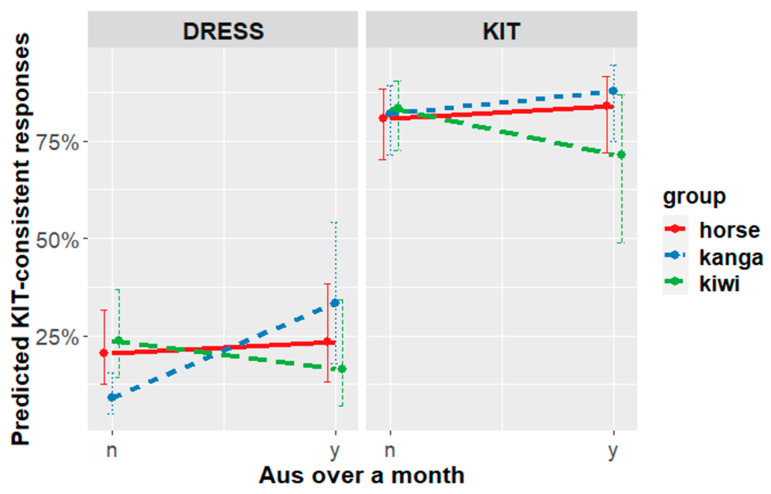
Experiment two model prediction plot from model fit to women only (model details in Appendix A). The figure shows a three-way interaction involving time spent in Australia (binary—on x-axis), prime-type (lines), and frame (panels).

**Table 1 brainsci-12-00684-t001:** Distribution of stimuli for experiment one across word-type for target items and fillers over three speakers.

**Frame Type**	**Example**	**Count**
KIT-frame	dXgging	45
DRESS-frame	sXnding	45
**Filler stimuli**	**Example**	**Count**
Real LOT	bothers	15
Fake LOT	fomments	15
Real STRUT	custom	15
Fake STRUT	duppet	15
Both-frame	bXgger	15
Neither-frame	kXzzard	15

**Table 2 brainsci-12-00684-t002:** Distribution of CELEX wordform frequencies (counts per 17.9 million) for KIT-frame and DRESS-frame stimuli.

	Minimum	Median	Max	Mean
KIT-frame	0.0	29.0	2270.0	161.7
DRESS-frame	0.0	26.0	827.0	81.78

**Table 3 brainsci-12-00684-t003:** Continua formant values (Hz) for the word ‘fixture’ at T-step 15 (median).

	F1	F2	F3
**NZ3 original *fixture***	568	1983	2879
** *Fixture* ** **synthesis 1**	568	1983	2879
** *Fixture* ** **synthesis 2**	531	2099	2871
** *Fixture* ** **synthesis 3**	495	2222	2864
** *Fixture* ** **synthesis 4**	460	2352	2856
** *Fixture* ** **synthesis 5**	426	2492	2849
**AU original *fixture***	426	2492	2849

**Table 4 brainsci-12-00684-t004:** Distribution of participants across conditions (number of men shown in parentheses).

	Speaking	Incidental	Total
**Baseline**	24 (6)	30 (12)	54
**Priming**	27 (5)	38 (12)	65
**Total**	51	68	119

**Table 5 brainsci-12-00684-t005:** Selected model for experiment one (dependent variable = KIT-consistent response. Intercepts = participant and word).

	Estimate	Std. Error	z Value	Pr (>|z|)
(Intercept)	−0.469	0.20714	−2.264	0.03
firstblock = yes	0.34007	0.07018	4.845	<0.0001
scaled order within block	−0.0691	0.03283	−2.105	0.04
frame = KIT	2.07562	0.21986	9.441	<0.0001
firstanimal = kanga	−0.2732	0.24416	−1.119	0.26
firstanimal = kiwi	−0.8441	0.23989	−3.518	<0.0001
frame = KIT: firstanimal = kanga	0.06594	0.16007	0.412	0.68
frame = KIT: firstanimal = kiwi	0.88564	0.16411	5.397	<0.0001

**Table 6 brainsci-12-00684-t006:** Distribution of participants across conditions. Number of men indicated in parentheses.

	Speaking	Incidental	Total (by Prime)
**Kangaroo**	18 (6)	28 (9)	46
**Kiwi**	20 (8)	21 (7)	41
**Horse**	26 (7)	23 (5)	49
**Total (by condition)**	64	72	136

**Table 7 brainsci-12-00684-t007:** Selected model for experiment two (dependent variable is KIT-consistent response. Intercepts are participant and word).

	Estimate	Std. Error	z Value	Pr (>|z|)
(Intercept)	−1.486276	0.249029	−5.968	<0.0001
primetype = kanga	0.132854	0.241932	0.549	0.58
Primetype = kiwi	0.2711	0.249015	1.089	0.28
scaled order	−0.33445	0.061424	−5.445	<0.0001
frame = KIT	2.836929	0.274642	10.33	<0.0001
prime = kanga: scaled order	0.290842	0.086805	3.351	<0.0001
prime = kiwi: scaled order	0.286136	0.089319	3.204	0.002
prime = kanga: frame = KIT	0.174694	0.130499	1.339	0.18
prime = kiwi: frame = KIT	−0.002354	0.132787	−0.018	0.99
scaled order x frame = KIT	0.028814	0.087952	0.328	0.74
prime = kanga: scaled order: frame = KIT	−0.075548	0.125966	−0.6	0.55
prime = kiwi: scaled order: frame = KIT	−0.388737	0.129347	−3.005	0.003

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
