# Peer review of "Social Priming in Speech Perception: Revisiting Kangaroo/Kiwi Priming in New Zealand English"

_brainsci, 2022, doi:10.3390/brainsci12060684_

Round 1

Reviewer 2 Report

This study further investigated the role of regionally-associated primes on speech perception in two experiments. The authors follow up on a previous finding using an updated methodology. Overall, I thought the study was very well motivated, clearly presented, and filled a gap of knowledge in the field.

I was very excited to review this paper, as I was big fan of the original stuffed animal study (I even got a stuffed Kiwi bird when I was in Australia who now sits on my shelf at work). It was a pleasure to read this paper and I only a have a few minor suggestions for that I hope will help strengthen the paper even more.

Comments:

1) I thought the authors had a very coherent and well-organized introduction. One thing I took issue with was that any many points throughout the introduction, the authors seemed to be suggesting that social information was influencing what people hear (e.g., line 42. It is still an open question whether social factors influence our perception or interpretation of speech. The former referring to what we hear, and the latter to what we think we hear.

E.g., see: 

Zheng, Y., Samuel, A.G. Does seeing an Asian face make speech sound more accented?. Atten Percept Psychophys 79, 1841–1859 (2017). https://doi.org/10.3758/s13414-017-1329-2

This is an important distinction and I don't think the authors have sufficiently demonstrated in their design that perception of the sounds was changed based on the priming. I would consider slightly modifying the word choices.

2) Do the authors think there is something inherently different about a stuffed animal in a waiting room as opposed to a cartoon animal on the screen? I was surprised not to see more discussion about this.

Firstly, there was a difference in where the participants were tested in virtual vs. in person situation. Also, the physical vs. virtual presence of the object.

Importantly, a stuffed animal has more social significance than an image. A stuffie is something that is owned. When participants see this prime in the waiting room, it has an inherent history to the owner (i.e., the lab) that has social significance.  I.e., the presence of a stuffed kangaroo in a waiting room is not arbitrary. For example, people typically interact with and purchase possessions that reflect themselves (e.g., Belk, 1988). Even very young children assume there is social significance to possessions and objects.

I don't this is an issue or anything like that, but I think it is important to note and discuss the differences in context this study as compared to its predecessor in the Discussion.

Minor comment: Line 134 sentence ends abruptly with a comma
